# Birth of a Healthy Monozygotic Twin Foal with Hydrops and a Dead Co-Twin

**DOI:** 10.3390/vetsci11120649

**Published:** 2024-12-13

**Authors:** Sofie Peere, Emma Van den Branden, Klaartje Broothaers, Ellen Polfliet, Katrien Smits, Jan Govaere

**Affiliations:** Department of Internal Medicine, Reproduction and Population Medicine, Ghent University, Salisburylaan 133, 9820 Merelbeke, Belgium; emma.vandenbranden@ugent.be (E.V.d.B.); klaartje.broothaers@ugent.be (K.B.); ellen.polfliet@ugent.be (E.P.); katrien.smits@ugent.be (K.S.); jan.govaere@ugent.be (J.G.)

**Keywords:** horse, pregnancy, hydrops, in vitro-produced embryo, twins

## Abstract

A 6-year-old multiparous standardbred recipient mare was presented at 9 months of gestation with a monozygotic twin foal with hydrops and a dead co-twin. Careful monitoring until 332 days of pregnancy resulted in the birth of one healthy, normal-sized foal accompanied with a non-macerated, non-mummified dead co-twin, diagnosed and corresponding to an 8-month-old fetus. This report outlines the clinical progression, treatment, positive outcome and prognosis of this compromised pregnancy.

## 1. Introduction

In vitro production (IVP) of equine embryos has become popular in the equine breeding industry. However, transfer of IVP equine embryos results in a higher prevalence of monozygotic twinning (1.6%; 4/254) compared to in vivo-derived embryos (0%; 0/413) [1]. The most common types of monozygotic twins in the mare are of monochorionic/diallantoic/diamniotic nature [1,2,3,4]. Due to the placental arrangement, diagnosis can only be made by ultrasound from visible embryo proper development (at 21 days), i.e., revealing two embryos within one embryonic vesicle [1,3,4,5]. If left untreated, the majority of cases result in spontaneous pregnancy loss of both conceptuses later in gestation [1]. Moreover, twin management techniques, commonly applied in dizygotic twins, are associated with unfavorable results in case of monozygotic twins, with only one successful case described after performing the thoracic compression technique [3,4]. As in humans, monochorionic monozygotic twins are expected to have intimate vascular anastomosis and placental sharing [1,3,6]. This suggests, that the demise of one fetus affects the viability of its co-twin, explaining the poor outcome of these pregnancies [1,4]. In women, survival of one twin is possible, both after spontaneous death or selective reduction of its co-twin with a monochorionic placenta [7,8]. However, these pregnancies are often classified as complex and high-risk with prematurity and stillbirth reported as adverse outcomes [7]. In commercial equine breeding systems, due to poor outcomes, termination of the monozygotic twin pregnancy before endometrial cup cell formation is the most appropriate treatment [3]. In addition, hydrops conditions are very exceptional in horses, with hydrallantois occurring more frequently than hydramnion [9]. Hydrops conditions are characterized by an excessive accumulation of fetal fluids (allantoic/amniotic), leading to an increased abdominal distention. Hydrallantois is typically reported to develop rapidly (within a few days) before clinical signs appear vs. hydramnion, which tends to progress more slowly [9,10]. The pathogenesis remains obscure but is associated with placental dysfunction (hydrallantois) and fetal abnormalities (hydramnion) [9,10]. Mares are often presented with an increased abdominal distention during mid- to late-term pregnancy, which may be accompanied by other clinical signs, such as depression, anorexia, colic, ventral edema, dyspnea and tachypnea [9,11,12]. Other than hydrops, the differential diagnosis for abnormal abdominal distention includes twin pregnancy and other causes of colic and ventral edema [9,11,13]. Diagnosis can be made by transrectal palpation, revealing a large distended uterus, bulging over the pelvic brim/into the pelvic cavity with an excessive amount of fetal fluid and no palpable fetus [9]. Furthermore, transrectal and abdominal ultrasonography enables assessment of fetal well-being, placental parameters, presence of twins and evaluation of the abdominal wall [14,15,16]. Hydrops conditions sometimes co-occur with twin (dizygotic) pregnancy, making it essential to pay attention to the presence of twins. Although, a direct causal link between the two conditions has not been established yet [11,12,17,18]. Additionally, abdominocentesis, followed by biochemical analysis, might help to determine the origin of fluid accumulation (amnio/allantoic) [12,19,20]. Possible complications associated with hydrops conditions include abdominal wall herniation, prepubic tendon rupture, uterine rupture, abortion, dystocia, delayed uterine involution, retained placenta and hypovolemic shock [10,11,13,21]. In fast progressing hydrallantoic cases, treatment mainly focuses on saving the mare, as most cases become clinically apparent at a time when the fetus is not viable (mid–late-term pregnancy) [10,13]. Given the slow progression of fluid accumulation and symptoms in case of hydramnion, careful monitoring until parturition and assistance during delivery may be considered [12]. As such, follow-up and treatment of hydrops conditions should be applied on a case-by-case basis, depending on the mare’s condition and the severity of clinical symptoms. In this report, the birth of a healthy monozygotic twin foal with hydrops and a dead co-twin is described.

## 2. Case Description

A 6-year-old multiparous standardbred recipient mare, 9 months pregnant (269 d), was referred carrying a monozygotic twin pregnancy, following transfer of a single IVP equine embryo. The presence of monozygotic twins (monochorionic, diamniotic, diallantoic) was diagnosed by ultrasound at 25 days (heartbeat check) and re-confirmed at 42 days of gestation. At that time, no twin management interventions were applied. However, upon referral, the mare was presented at the clinic for evaluation of the pregnancy, fetal viability and further advice and follow-up until parturition. On presentation, the mare appeared bright, did not exhibit any signs of discomfort, and had no appreciable mammary gland development. During rectal palpation, the uterus was distended over the pelvic brim, and no fetuses were readily palpable. On transrectal ultrasonography, an excessive amount of fetal fluid was observed. However, the origin (amniotic or allantoic) of the fluid was difficult to distinguish. Combined thickness of the uterus and placenta (CTUP) measurements were within reference limits (<7 mm) [22,23,24], and the cervix was closed. Transabdominal ultrasonography revealed the presence of twins in the cranial abdomen. One fetus was in anterior presentation with a clear heart rate of 96 bpm. The other fetus was diagnosed in posterior presentation, without a detectable heart rate, presuming death of the fetus. Based on the clinical history, symptoms, and examinations, the presumed diagnosis—pregnancy with monozygotic twins (monochorionic diallantoic diamnionic) in presence of hydrops and death of one fetus—and its possible complications, especially the potential disastrous consequences of twin pregnancy [3] and risks of hydrops in the mare [9], were discussed with the owner. The owner was instructed to report any occurrence of abnormal clinical symptoms, increased abdominal distention and premature mammary gland enlargement in the mare. Two weeks before due date (at 320 days of pregnancy), the mare was re-hospitalized at the clinic for an assisted parturition. At examination, the mare exhibited an increased abdominal distention and moderate amount of ventral edema; however, no signs of discomfort were present. During rectal palpation, the uterus was even more distended, had a tense turgor and bulged out above the pelvic brim without palpable fetuses. Ultrasonography of the abdominal wall revealed no trauma to the abdominal muscles or prepubic tendon, and no signs of pain were present when palpating the abdominal flank. Ultrasonography re-confirmed the presence of excessive fetal fluid, one live fetus in anterior presentation with a heart rate of 72 bpm and one dead fetus in posterior presentation. Amnio-/allantocentesis was not performed, because of the known potential risks of the procedure [25], and a presumed diagnosis of hydramnion vs. hydrallantois was made based on the clinical history [9]. To support the abdominal wall, an abdominal bandage was placed. The mare was placed under continuous surveillance for signs of discomfort, abdominal distention, mammary gland enlargement and Brix index. The mare was evaluated every other day by ultrasound, and her clinical parameters, body measurements and bodyweight were closely monitored. At 332 d of pregnancy, the mare had an assisted vaginal delivery. Moderate manual traction was applied, and the living foal in anterior presentation dorsosacral position, was delivered within 15 min after the start of stage II. Herewith, the mare expelled a huge amount of amniotic fluid. Subsequently, a second foal was palpated in posterior presentation, dorsosacral position and breech posture. The second foal was dead, and extraction was accomplished after mutation. At the gross examination, the foal was small (12 kg; withers height: 56 cm; crown-rump length: 60 cm), fresh, hairless and non-autolyzed, corresponding to an 8-month-old fetus [26,27] (Figure 1).

Shortly after delivery, the mare exhibited clinical signs of hypovolemic shock (tachycardia, light dyspnea and instable) and shock therapy was initiated. A single bolus of 2 L hypertonic sodium chloride solution (7.5% NaCl; IV; B. Braun) and 0.06 mg/kg BW Rapidexon (2 mg/mL dexamethasone sodium phosphate; IV; Dechra Regulatory BV) were administered. The mare recovered quickly, and further administration of maintenance infusion (isotonic, ringer-lactate; IV; Baxter) together with monitoring of clinical (heart-/and respiratory rate, mucosae, capillary filling time and body temperature) and blood parameters (blood gas analyses: hematocrit (Hct), base excess (BE), pH, lactate, glucose, PCo2, Po2 and electrolytes) were continued until normalization. The live born colt was considered healthy, normal-sized (40 kg; withers height: 94 cm; crown-rump length: 90 cm) and showed normal Appearance, Pulse, Grimace, Activity, Respiration (APGAR) score and physical activity (suckling reflex, first standing, and meconium discharge) [28,29]. Antibodies were tested (SNAP Foal IgG test; Elisa test; IDEXX) 18 h post-partum, and a general blood examination (CBC; BE, pH, lactate, glucose, PCo2, Po2 and electrolytes) was performed, whereas all values were within normal limits. A 7-day follow-up post-parturition was performed at the clinic. During the foal’s hospitalization, no pathological changes in clinical appearance and/or blood parameters were observed, concluding the newborn colt to be healthy. Fetal membranes were expelled (Stage III) within 120 min after foaling. A thorough examination of the placenta was conducted and revealed two separate amnions and one chorion. A clear fusion of the umbilical cords was noticed at the site where the allantoic membranes meet (Figure 2), concluding the presence of monochorionic/diallantoic/diamniotic monozygotic twins. The amnion of the live born foal was found to be enlarged, as confirmed by fetal membrane measurements (surface amnion: 2.4 m^2^). Macroscopically, no inflammatory or pathological changes were observed. Placental histological examination did not reveal any abnormalities.

Before and after foaling, the mare weighed 698 kg and 550 kg, respectively. The total weight of the twins (live and dead: 52 kg) and placenta (8.6 kg) contained 60.6 kg. As such, a loss of 87.4 kg (resp. 87.4 L) of fluid was estimated. Acknowledging the normal amount of fetal fluids (3–5 L amniotic-and 8–15 L allantoic fluid) in a near-term light breed pregnant mare [3], hydrops was present in this case. The amount of fetal fluids expelled from each cavity, the amniotic/allantoic cavity, respectively, could not be determined; however, the presumed diagnosis of hydramnion was based on the clinical and fetal membrane findings. Post-partum rectal palpation and ultrasonographic follow-up of the mare’s uterus revealed a large, atonic uterus with a moderate amount of hyperechoic fluid. Uterine lavages were performed twice daily (10–20 L of 0.9% NaCl), combined with oxytocin (10–20 IU; IM; q 4–6 h; Kela NV) administration. Treatment was discontinued after 4 days, when the uterus had decreased in size, intra-uterine fluid accumulation was minimal and contractility had improved. Both mare and foal were discharged from the clinic one week after parturition. Four months after discharge, the foal and mare were healthy, the foal exhibited a normal growth rate and was similar in size to peers.

## 3. Discussion

In the current case, monozygotic twin pregnancy was evident. However, confirming twin pregnancy during late gestation by ultrasonography can be quite challenging [14]. The presence of other clinical signs like abdominal distention and premature mammary gland development with/without lactation could aid in making a conclusive diagnosis [22,30]. Premature mammary gland development is a common clinical sign of impending abortion and is often seen in dizygotic twins after death of one or both of the fetuses, due to placental insufficiency [22,31]. While fetal death in one member of the twin was confirmed by ultrasound at 9 months of gestation here, no udder distention nor lactation was present. It seems important to bear in mind that these clinical signs, indicating impending abortion may not be present in cases of monozygotic twin pregnancies after death of one of the twins in contrast to what is mostly seen in dizygotic twin pregnancies. Here, considering the negative consequences of twin pregnancy [3] and the poor prognosis of foal survival and health of the mare, due to hydrops conditions [9,12], the safest and most recommended alternative treatment would have been to induce abortion at 9 months of gestation, when the mare was first presented. However, previous reports described foals surviving from a hydramnion [12], in which close monitoring and follow-up until parturition were the key to success. In case of high-risk pregnancies, attendance at parturition is essential to address potential complications. Surveillance was maintained with monitoring clinical signs of impending parturition and follow-up of Brix-index (%) in the mammary gland secretions. At 332 days of gestation, the mare had an assisted vaginal delivery and gave birth to one dead foal and one healthy, normal-sized, live foal. Treatment of hypovolemic shock and a thorough genital tract evaluation were performed, as recommended in the presence of hydrops and dystocia in the mare [9,32]. The recovery was uncomplicated, and no further abnormalities were reported. As with any twin pregnancy in the horse, the limited placental area from which two fetuses can derive sufficient nutrition to bring them to term makes the live birth unlikely [33]. So far, no previously reported cases described the birth of a live foal after spontaneous intra-uterine death of its co-twin with a monochorionic placenta. Expecting these type of twins to have a shared placental environment (chorion), it remains unclear how the death of the fetus did not affect the viability of its co-twin in the current case. Another unusual characteristic was the fresh, non-autolyzed state of the dead foal, despite the expectation of autolysis or mummification following confirmed fetal death at 9 months of gestation. The shared chorion in monochorionic monozygotic twins is further perfused and may have restrained the cascade thought to be responsible for these processes. In humans, ‘twin reversed arterial perfusion’ (TRAP) sequence is a complication described in monochorionic twins, characterized by a non-viable, acardiac fetus accompanied with multiple anomalies and a ‘pump’ fetus that feeds the acardiac fetus through vascular anastomosis in the placenta [6,34]. This phenomenon could explain continued perfusion of the dead co-twin in this case. However, TRAP is associated with other anomalies and a poor survival rate of the ‘pump’ fetus by heart failure and prematurity [34], which were not observed here. Additionally, considering the differences in placental structure between humans and horses, it is not clear whether this extrapolation holds. With the more frequent publication of reports mentioning monochorionic twin gestation in the mare, it may be worth investigating the nature of placental sharing, type of vascular anastomosis and degree of placental insufficiency and their impact on fetal outcome. Given the risks associated with the birth of dysmature foals as sequalae of twin pregnancy [3], a comprehensive follow-up is essential to diagnose and/or rule out the presence of complications. In the current case, the foal exhibited no signs of dysmaturity or weakness at birth, and a 7-day follow-up did not reveal any abnormalities, concluding the neonate to be healthy. Both mare and foal were discharged from the clinic one week after parturition. Four months after discharge, the mare and foal were reportedly doing well. The foal was healthy, exhibited a normal growth rate and was similar in size, compared to peers.

## 4. Conclusions

To the author’s knowledge, this is the first report of a healthy foal born from a monozygotic twin pregnancy in presence of hydrops and a dead co-twin. However, it needs to be highlighted that this is an exceptional case. The most rational advice remains that a monozygotic twin pregnancy is better terminated and that, in cases of hydrops, either inducing abortion or careful monitoring until parturition may yield a positive outcome. Nonetheless, each case should be approached on an individual basis.

## Figures and Tables

**Figure 1 vetsci-11-00649-f001:**
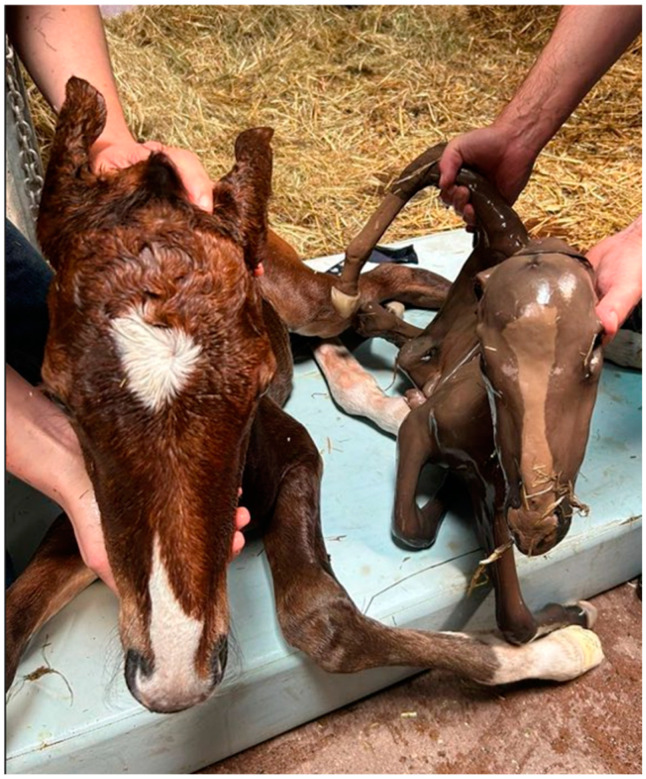
Healthy, normal-sized foal (left) and dead co-twin (right) derived from a monozygotic twin pregnancy with hydrops (Ghent University).

**Figure 2 vetsci-11-00649-f002:**
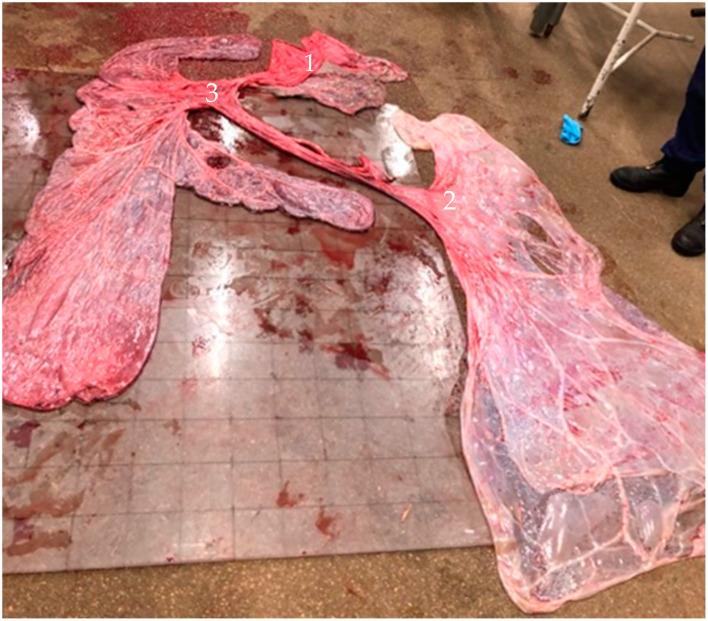
Placenta derived from a monozygotic twin pregnancy with hydrops and a dead co-twin: two separate amnions (1—dead foal; and 2—live born foal, enlarged amnion); one chorion with a clear fusion of the umbilical cords at the site where the allantoic membranes fuse (3) (Ghent University).

## Data Availability

Data are unavailable due to privacy restrictions.

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
