# Peer review of "Birth of a Healthy Monozygotic Twin Foal with Hydrops and a Dead Co-Twin"

_vetsci, 2024, doi:10.3390/vetsci11120649_

Round 1

Reviewer 1 Report

Comments and Suggestions for Authors

- line 9 higher incidence, line 26 higher prevalence :higher than in vivo produced embryos? Please add and provide the number of this data

- line 74-75 please explain why no intervention was applied, this is quite surprising

- line 102 diagnosis of hydramnion has been already performed..(see line 88): please clarify and explain how hydrammnion diagnosis was done in line 88

Author Response

Reviewer 1:

Comment 1 :

-line 9 higher incidence, line 26 higher prevalence :higher than in vivo produced embryos? Please add and provide the number of this data

Line 26-->Lines 32-33: I’ve added the number of the data (reported in reference [1]) and changed the sentence to:

‘However, transfer of IVP equine embryos results in a higher prevalence of monozygotic twinning (1.6%; 4/254) compared to in vivo derived embryos (0%; 0/413).’

Reference [1]:

Dijkstra, A.; Cuervo-Arango, J.; Stout, T.A.E.; Claes, A. Monozygotic Multiple Pregnancies after Transfer of Single in Vitro Produced Equine Embryos. Equine Vet J 2020, 52, 258–261, doi:10.1111/evj.13146.

 Comment 2:

- line 74-75 please explain why no intervention was applied, this is quite surprising

Lines 74-75: I completely agree with your question/statement. On demand of the owner, because of high genetic value of the offspring,  no twin management interventions were applied. However, in my opinion, termination of the pregnancy < 35 days of gestation (before endometrial cups formation) would have been the treatment of choice, as I comment in the manuscript.

Furthermore, one could have tried to perform ‘the thoracic compression technique’  (55-65 days of gestation) to reduce one of the two fetuses, as one successful case was yet described ([4]: Peere et al., 2022). However, because of the low success rate, this technique would not be the preferred treatment of choice.

Reference [4]:

Peere, S.; Papas, M.; Gerits, I.; Van den Branden, E.; Smits, K.; Govaere, J. Management of Monozygotic Twins in the Mare. J Equine Vet Sci 2022, 113, 103988.

 Comment 3:

- line 102 diagnosis of hydramnion has been already performed..(see line 88): please clarify and explain how hydrammnion diagnosis was done in line 88

I completely agree with your comment. I’ve corrected and clarified the sentences in the manuscript. Although, the diagnosis of ‘hydrops’ was conclusive on the first presentation, distinguishing the presence of hydramnion vs. hydrallantois was not possible until the known clinical history and placental findings. As such, I’ve changed some sections in the manuscript:

Lines 51-56: I’ve added these sentences to explain better the difference between hydrallantois vs. hydramnion.

Line 88--> 98-102: ‘’ Based on the clinical history, symptoms, and examinations, the presumed diagnosis - pregnancy with monozygotic twins (monochorionic diallantoic diamnionic) in presence of hydrops and death of one fetus - along with its possible complications, especially the potential disastrous consequences of twin pregnancy [3] and risks of hydrops in the mare [9], were discussed with the owner.’’

Line 102-->114-115: Amnio-/allantocentesis was not performed, because of the known potential risks of the procedure [25] and a presumed diagnosis of hydramnion vs. hydrallantois was based on the clinical history [9].

Lines 114-115:

Explanation:

At the first examination: rectal palpation and transabdominal ultrasonography revealed a distended uterus without palpable fetuses and an excessive amount of fetal fluid. A such, hydrops was present in this case, however the origin of the fetal fluid could not be determined (no amnio-/allantocentesis was performed). In addition, the mare appeared bright, did not show any signs of discomfort or abnormal abdominal distention.

When the mare was re-hospitalized and -examined, a progression was noticed in the amount of fetal fluid, however the mare’s condition did not deteriorate during pregnancy, defining the clinical progression as ‘slow’ and a presumed diagnosis of hydramnion was made.

In contrast, hydrallantoic cases are characterized by a fast clinical progression (few days), signs of discomfort, abnormal abdominal distention and a detrimental outcome, when left untreated. Termination of the pregnancy (induction of abortion by slow drainage through a cervical catheter), should therefore always be attended. In addition, placental abnormalities are often detected/related to the presence of hydrallantois, which were not observed in the current case.

Reviewer 2 Report

Comments and Suggestions for Authors

This clinical case represents a notable and substantial contribution to the field of equine reproduction science. The rarity of monozygotic twin gestations in mares, combined with the unusual presentation of hydramnion and the demise of one of the twins, underscores the novelty of this case, also provides useful information on the clinical management of hydramnios and monozygotic twin gestations in equines. Furthermore, the successful birth of a healthy foal from a monozygotic twin gestation following the transfer of a single in vitro-produced (IVP) embryo highlights the importance of close monitoring and follow-up during pregnancies achieved through these reproductive techniques. Despite significant advancements, these techniques still pose unique challenges, including the high prevalence of monozygotic twinning, which is often associated with poor gestational outcomes. The exceptional circumstances described in this report not only expand the current knowledge base but also emphasize the need for further research and innovation in managing the complex equine pregnancies associated with IVP. This need is becoming increasingly urgent due to the growing commercial use of IVP technologies in recent years.

Line 35 change "human" to "humans"

Author Response

Thank you very much for considering this manuscript to be published. Thank you for the positive comment and I’m delighted to present novel insights regarding managing ‘high risk’ pregnancies in the mare and extending our knowledge on these topics. we have found it a very interesting case ourselves and hope this manuscript will serve as a valuable contribution to the field of equine reproduction research and practice.

Line 35 change "human" to "humans"

Thank you for the suggestion,

Line 41: I’ve changed the word.

Reviewer 3 Report

Comments and Suggestions for Authors

This case report describes an interesting case with a positive outcome that was never observed before. The case is well explained. I agree with the authors that terminating the pregnancy is the most rational advice for a monozygotic twin pregnancy. However, this article provides new insights to the practitioners working in the equine reproduction field.

Author Response

Thank you very much for the positive comment and considering the manuscript to be published!

Reviewer 4 Report

Comments and Suggestions for Authors

The study presents a rare and significant case of a monozygotic twin pregnancy in a mare complicated by hydramnion. The case addressed highlights the complexities and risks involved in such pregnancies, which are generally associated with poor outcomes.

Clinical Significance:
The findings underscore the importance of thorough ultrasound assessments in early gestation to identify twin pregnancies, which can be easily overlooked. The report notes that while monozygotic twins are relatively rare, their incidence appears heightened in the context of IVP embryos. Additionally, the prognosis for such pregnancies can be dire. This is the first reported case of a healthy foal that was delivered despite the presence of a dead monozygotic twin and hydramnion.

This case report serves as a valuable contribution to equine obstetrics literature.

Author Response

(The authors gave the same response as above.)

Reviewer 5 Report

Comments and Suggestions for Authors

This is a well written case report. It provides relevant information on a twin pregnancy and does not require any changes. Well done!

Author Response

(The authors gave the same response as above.)

Reviewer 6 Report

Comments and Suggestions for Authors

Overall, this is a unique and intriguing clinical case that will be of interest to practitioners. However, the case presentation could benefit from a more detailed description and in-depth discussion. 

Comments on the Quality of English Language

The language could be improved for consistency and formality, as some sections are too informal, and certain features are described superficially

Author Response

Overall, this is a unique and intriguing clinical case that will be of interest to practitioners. However, the case presentation could benefit from a more detailed description and in-depth discussion. Additionally, the language could be improved for consistency and formality, as some sections are too informal, and certain features are described superficially.

Thank you for your comment. I’ve changed and adapted the manuscript with the aid of your suggestions, making it more detailed and constructive for the reader. I agree my English language could be improved in some sections throughout the manuscript, especially with the focus point on grammar. A native Englisch speaker at Ghent University has checked the manuscript before re-submission.

Comment 1:

Title: in my opinion, you should remove “amnion” because the diagnosis of hydrops was certain, the diagnosis of hydrops amnion turns out to be highly probable. 

I completely agree with you, I’ve changed the title. The diagnosis of hydramnion was indeed highly probable and not strict conclusive. However, the clinical history (slow progression of the condition, bright appearance of the mare, no abnormal clinical signs) and fetal membranes findings (enlarged amnion; surface amnion: 2,4 m2 ) indicates the presence of hydramnion vs. hydrallantois.

In addition, as a mention at line 166: it’s a presumed diagnosis of hydramnion.

Comment 2:

Abstract

Lines 8-9: please, rephrase the sentence.  Your intention seems to be highlighting the contrast between the literature and clinical experience, but the current phrasing comes across as contradictory.

Lines 14-15

I agree that the sentence could come across as contradictory. However, we do need to acknowledge that the presence of monozygotic twins in the mare remains rare, especially compared to the prevalence of dizygotic twins. I’ve added the number of data at lines 32-33 to highlight the prevalence of these cases.

Reference [1]:

Dijkstra, A.; Cuervo-Arango, J.; Stout, T.A.E.; Claes, A. Monozygotic Multiple Pregnancies after Transfer of Single in Vitro Produced Equine Embryos. Equine Vet J 2020, 52, 258–261, doi:10.1111/evj.13146.

Comment 3 :

Line 13: “diagnosed with” instead of “diagnosed of”.

Line 19

Thank you for the correction, I’ve changed the word.

 Comment 4:

Line 19: please, delete "Extremely, since "unique" is an absolute term that doesn't need modifiers.

Line 25:

Thank you for the suggestion, I’ve deleted the word.

Comment 5:

Introduction

Lines 28-30: do you mean “from 21 days post IVP”? in this case rephrase the sentence please, i.e. "Due to the placental arrangement, diagnosis can only be made by ultrasound starting from 21 days post IVP, when proper embryo development becomes visible, revealing two embryos within a single embryonic vesicle".

Lines 35-36:

Thank you for the suggestion. I agree, this sentence can be a little bit confusing, however it indicates that the monozygotic twin pregnancy (monochorionic, diallantoic, diamniotic) can only be visualized at embryo proper development and onwards (revealing two embryos within a single embryonic vesicle). In normal pregnancy (post-AI; natural covering; embryo-transfer in-vivo embryos), this occurs at 21 days post-ovulation. However, because of the slightly slower progression of ICSI embryos during pregnancy in contrast with in-vivo/normal pregnancy, we cannot always assume that this is strictly at 21 days post IVP. Furthermore, each cryopreserved blastocyst can be of different age/speed of development (6-10 days post ICSI), meaning that this information should be available on the moment of embryo-transfer, which is mostly not the case. In conclusion, monozygotic twin pregnancy (monochorionic, diallantoic, diamniotic) can only be diagnosed if the pregnancy resembles this stage (21-25 days post-ovulation) on ultrasound.

Comment 6:

Line 49: “dyspnea, tachypnea” instead of “dyspnee, tachypnee”. Please, remove “etc.”

Line 59:

Thank you for the grammatical correction, I’ve changed the words.

Comment 7:

Line 67: could you please provide a clearer explanation of why the description of this case would be useful for practitioners, specifically highlighting the significance of this case report, which currently appears only implicitly?

Lines 78-80:

Thank you for your comment. As transfer of IVP embryos is increasing in the recent years, this case report is important/useful for practitioners to be aware of the occurrence of these monozygotic twins. Especially, when checking the mare by ultrasound for pregnancy, the monozygotic twin will only be visualized on the ‘second’ control for heartbeat (25 days of gestation). Thereafter, if a twin pregnancy is present, a management/decision plan should be discussed with the owner, understanding the poor outcomes of these pregnancies. In addition, managing high risk pregnancies in the mare is an important task in performing stud- and obstetrics medicine. The practitioner should know, understand and recognize clinical symptoms related to ‘high risk’ conditions in the pregnant mare with its negative implications on both mare and fetus/foal. Hence, the practitioner can decide to manage/treat the condition or refer the pregnant mare to a clinic for a second opinion/to follow-up.

Comment 8:

Lines 73-75: if you can, please add the picture of the ultrasound examination, showing the monochorionic, diamniotic, diallantoic condition. Otherwise, explain, even with a drawing, which characteristics of the ultrasound examination led to this diagnosis.

Lines 84-86:

I agree and understand that an additional ultrasound image/drawing would be beneficial to the manuscript, however multiple references are mentioned, explaining the placental nature of these monozygotic twins (monochorionic; diallantoic; diamniotic). As being restricted to a maximum word/figure limit in the case report, I only described this topic briefly in the manuscript.

References explaining the placental nature of monozygotic twins:

-Peere, S.; van Den Branden, E.; Papas, M.; Gerits, I.; Smits, K.; Govaere, J. Twin Management in the Mare: A Review. Equine Vet J 2024, 56, 650–659, doi:10.1111/evj.14094. (figure included)

- Dijkstra, A.; Cuervo-Arango, J.; Stout, T.A.E.; Claes, A. Monozygotic Multiple Pregnancies after Transfer of Single in Vitro Produced Equine Embryos. Equine Vet J 2020, 52, 258–261, doi:10.1111/evj.13146.

- Mancill, S.S.; Blodgett, G.; Arnott, R.J.; Alvarenga, M.; Love, C.C.; Hinrichs, K. Description and Genetic Analysis of Three Sets of Monozygotic Twins Resulting from Transfers of Single Embryos to Recipient Mares. J Am Vet Med Assoc 2011, 238, 1040–1043, doi:10.2460/javma.238.8.1040. (figure included)

Comment 9:

Lines 86-90: please, check the punctuation and rephrase, i.e.: "Based on the clinical history, symptoms, and examinations, the presumed diagnosis — pregnancy with monozygotic twins (monochorionic, diallantoic, diamniotic) in the presence of hydramnion and the death of the second fetus — along with its possible complications, especially the potential disastrous consequences of twin pregnancy [3] and the risks of hydramnion in the mare [9], were discussed with the owner."

Lines 98-102:

Thank you for the suggestion, I agree the sentence was not clear. Additionally, I’ve changed ‘hydramnion’ in ‘hydrops’

Comment 10:

Line 91: “detect or report” instead of “follow-up”? Symptoms, “such as”….

Line 103:

Thank you for the suggestion, I’ve changed the word.

Comment 11:

Line 92: please, explain what you mean by “due date” and how you determine it.

Thank you for your comment.

As we do know, the physiological gestation length is variable and ranges from 320 to 360 days in the mare. Mostly we recommend the client to predict the parturition depending on the gestation length (depending on the ovulation/embryo-transfer date; 335-345 days) and the presence of clinical signs, indicating impending parturition (mammary gland development, …).

As a clinician, several other parameters can be used to determine readiness to foal (‘due date’), including: pH (<6.4) , Conductivity (<4.8 mS), an increase in Brix index (>20%) and CalciumCarbonate (CaCo3; >250 ppm) in the mammary gland secretions (MGS).

References:

-  Nagel, C.; Aurich, C. Induction of Parturition in Horses – from Physiological Pathways to Clinical Applications. Domest Anim Endocrinol 2022, 78, doi:10.1016/j.domaniend.2021.106670.

- Magalhaes, H.B.; Colombo, I.; Spencer, K.M.; Podico, G.; Canisso, I.F. Conductivity of Mammary Gland Secretions Is a Sensitive and Specific Predictor of Parturition in Mares. Equine Vet J 2024, 56, 719–725, doi:10.1111/evj.14070.

 Comment 12:

Line 100: please, replace “heart rate” with “heart rate” here and throughout the text. Reference n. [16] should be removed as only clinical data are reported. The meaning of heart rate should be commented on in the discussion. 

Line 112:

Thank you for the grammatical correction. I’ve changed the word throughout the text and also removed the reference n. [16].

Comment 13:

Lines 102-103: please, indicate here and comment in discussion which clinical and ultrasonographic parameters allowed the differential diagnosis between hydramnios and hydrallantois.

Thank you for your comment. I agree with you that in the manuscript it’s not clearly mentioned how the differentiation between hydramnion vs. hydrallantois was made. However, as you mention in your first comment, the diagnosis of hydrops was conclusive, however the presence of hydramnion was highly probable. I clarified this topic throughout the manuscript. In addition, the presumed diagnosis of hydramnion was mainly based on the clinical course of  the case and the fetal membranes findings (lines 114-115)

Explanation:

At the first examination: rectal palpation and transabdominal ultrasonography revealed a distended uterus without palpable fetuses and an excessive amount of fetal fluid. A such, hydrops was present in this case, however the origin of the fetal fluid could not be determined (no amnio-/allantocentesis was performed). In addition, the mare appeared bright, did not show any signs of discomfort or abnormal abdominal distention. When the mare was re-hospitalized and -examined, a progression was noticed in the amount of fetal fluid, however the mare’s condition did not deteriorate during pregnancy, defining the clinical progression as ‘slow’ and a presumed diagnosis of hydramnion was made. In contrast, hydrallantoic cases are characterized by a fast clinical progression (few days), signs of discomfort, abnormal abdominal distention and a detrimental outcome, when left untreated. Termination of the pregnancy (induction of abortion by slow drainage through a cervical catheter), should therefore always be attended. In addition, placental abnormalities are often detected/related to the presence of hydrallantois, which were not observed in the current case.

Comment 14:

Line 113: please, add “at the gross examination, the foal….”

Lines 124-125: Thank you for the suggestion, I’ve changed it in the manuscript.

Comment 15:

Lines 118-120: please specify that you are referring to the mare. Additionally, could you clarify what you mean by 'instable'? Are you suggesting that the mare is unable to maintain her stance, or that she is wobbling? The treatment should be better described: the hypertonic solution was administered as single bolus? What about the maintenance? Specific data of the dexamethasone, such as commercial name, company, producer data? Were tested blood parameters other than blood-gas analysis? Can you add e table with the blood data analysis.

Thank you for the suggestions.

Lines 127-129: I’ve changed the sentence indicating clearly that I’m referring to the mare. In addition, I’ve added the specific data (commercial name, company) of the medication that was administered, the maintenance infusion after the initial shock therapy and described more clearly which clinical-and blood parameters that were monitored.

As I’m restricted to a maximum word/figure limit in the manuscript, I will not be able to share an additional table showing the exact blood parameters of the mare. In addition, In my opinion this data will not change the treatment/outcome described in this case. However, if the reviewer strongly requests this, I can still send these blood parameters personally to the reviewer. 

Comment 16:

Lines 122-123: please, specify the APGAR score of the foal and what you mean by “normal physical activity”.

Thank you for the suggestions.

  • Line 136: I’ve changed and clarified the abbreviation ‘APGAR’ in the sentence as: Appearance, Pulse, Grimace, Activity, Respiration (APGAR) score.
  • Line 137: By normal physical activity I refer to the first time standing, suckling reflex and meconium discharge. All these parameters were within normal values and are always checked within the first hours after the foal’s birth.

Comment 17:

Line 123: please, add the data of the Elisa test (company, producer, …).

Thank you for the suggestion, I’ve added the missing information regarding the ELISA test (lines 137-138).

Comment 18:

Lines 133-134: please, add further information about the histological examination: stains, techniques,

findings, criteria of evaluation.

Lines 154-155:

Thank you for the suggestion. I understand that a more detailed report/description of the histological examination of the fetal membranes would be valuable. However, no abnormalities were found on histological examination. As we do know, hydrops conditions, especially hydrallantois, are frequently related/caused by placental dysfunction, in which we detect fetal membranes abnormalities on both macroscopic and histological examination (analyzed samples in this case: different sites of allanto-amnion (both dead and live foal); allantochorion), suggesting the presence of hydramnion in this case. As sample tissues are always sent and analyzed by the pathologist, we receive a final report to highlight any normal/abnormal findings, which could aid in formulating a correct and conclusive diagnosis. In my opinion, this seems to be sufficient for the reader within the scope of the manuscript.

Comment 19:

Line 141: please, delete comma after “Acknowledging”.

Line 162:

Thank you for the correction, I’ve deleted the comma.

Line 145: please, replace “placental findings” with “fetal membranes’ findings”.

Line 166:

Thank you for the suggestion, I’ve changed the word.

Comment 20:

Discussion

In my opinion, an in-depth discussion should be included on the following topics: the differential diagnosis between hydroamnios and hydroallantois; the formation of two amnia, two allantoids, and one chorion during fetal and membrane development; the pros and cons of not inducing abortion during the first visit at 9 months; and udder behavior, including the absence of udder enlargement and premature lactation.

Thank for your suggestion. I have explained the difference between hydramnion and- allantois more in depth in the introduction (lines: 51-57). In the discussion, I commented on the clinical signs, indicating impending abortion in the mare. Furthermore, I agree that there are more topics to be discussed regarding this case, however being restricted to a maximum limit of words, I’ve highlighted the most important key-points/interesting features. However, I agree that more investigation/research should be done in the field of these ‘high risk’ pregnancies.

Comment 21:

Lines 155-156: please rephrase the sentence, i.e. delete “yet” and divide the sentence: “In the current case, monozygotic twin pregnancy was evident. However, confirming twin pregnancy during late gestation by ultrasonography can be quite challenging”.

Lines 176-177:

Thank you for the suggestion. I’ve changed it in the manuscript.

Comment 22:

Line 162: please, add here comments about the absence of udder enlargement and premature lactation, eliminating the sentence at lines 206-209 in the “Conclusion”.

Thank you for your suggestion. I’ve deleted the section at the conclusion and included the comment in the discussion (lines 184-186).

Reviewer 7 Report

Comments and Suggestions for Authors

The manuscript titled ‘Birth of a healthy monozygotic twin foal with hydrops amnion and a dead co-twin’ describes the only known case of hydrops in a mare that resulted in the birth of monozygotic twin foals, with one foal born alive and healthy and the other corresponding to a foal at 8 months of gestation. This is a great case study and importantly highlights gaps in foal and mare management in hydramnion cases involving monozygotic twin foals.

Specific comments:

L115, 130 – please replace the reference to picture with the word figure.

L116 – there is no figure caption. Please update this for figure 1.

L117-119 – This sentence should have a closed parenthesis at the end of the sentence following the treatments. Otherwise, this sentence suggests there is more to follow.

L118 - .. should be replaced with other clinical signs if present, otherwise please remove.

L119-121 – please describe the parameters that were monitored. You also mention blood-gas, but this is also non-descriptive.

L123-125 – please refer to the above comment. Please describe the examinations performed.

L133-134 – you mention histopathology of the placenta, but can you please explain how many sites you examined and whether these sites pertained to both twin foals, or just one. Can you please also describe whether you histopathologically examined both amnions in addition to the chorion and umbilical cords? If so, please include these results in the discussion.

L176, 177 – please change the word dead to death.

Comments on the Quality of English Language

For the most part, the English language throughout the manuscript is good, but in some areas, this can be improved for improved clarity and readability. Additionally, grammar and punctuation could also be improved.

Author Response

Specific comments:

Comment 1 :

L115, 130 – please replace the reference to picture with the word figure.

Thank you for the suggestion,

Lines 127, 151: I’ve changed picture to figure (Figure 1 and Figure 2).

Comment 2:

L116 – there is no figure caption. Please update this for figure 1.

I agree, thank you for noticing.

Lines 129-130: I’ve added the figure caption and also changed ‘picture’ to ‘figure’.

Comment 3:

L117-119 – This sentence should have a closed parenthesis at the end of the sentence following the treatments. Otherwise, this sentence suggests there is more to follow.

Thank you for the comment.

Lines 132-140: I’ve changed the structure of the sentences and included additional relevant information regarding the treatment of the case presented.

Comment 4:

L118 - .. should be replaced with other clinical signs if present, otherwise please remove.

I agree, I removed it (lines 132-133).

Comment 5:

L119-121 – please describe the parameters that were monitored. You also mention blood-gas, but this is also non-descriptive.

I agree, thank you for the suggestion.

Lines 137-140: I’ve described the clinical- and blood parameters that were monitored.

Comment 6:

L123-125 – please refer to the above comment. Please describe the examinations performed.

I agree, thank you for the suggestion.

Lines 137-140: I’ve added the performed examinations.

Comment 7:

L133-134 – you mention histopathology of the placenta, but can you please explain how many sites you examined and whether these sites pertained to both twin foals, or just one. Can you please also describe whether you histopathologically examined both amnions in addition to the chorion and umbilical cords? If so, please include these results in the discussion.

Lines 154-155:

Thank you for the suggestion. I understand that a more detailed report/description of the histological examination of the fetal membranes would be valuable. However, no abnormalities were found on histological examination. As we do know, hydrops conditions, especially hydrallantois, are frequently related/caused by placental dysfunction, in which we can detect fetal membranes abnormalities on both macroscopic and histological examination (analyzed samples in this case: different sites of allanto-amnion (both dead and live foal); allantochorion). However, this was not the case here. As sample tissues are always sent and analysed by the pathologist, we receive a final report to highlight any normal/abnormal findings, which could aid in formulating a correct and conclusive diagnosis. In my opinion, this seems to be sufficient for the reader within the scope of the manuscript.

Comment 8:

L176, 177 – please change the word dead to death.

Indeed, thank you for correcting. I’ve changed the word (line 177).

Comment 9:

For the most part, the English language throughout the manuscript is good, but in some areas, this can be improved for improved clarity and readability. Additionally, grammar and punctuation could also be improved.

I agree my English language could be improved in some sections throughout the manuscript, especially with the focus point on grammar. A native Englisch speaker at Ghent University has checked the manuscript before re-submission. Thank you for the suggestion.

Round 2

Reviewer 6 Report

Comments and Suggestions for Authors

Thank you for resubmitting and making substantial changes to your manuscript.Many improvements are evident. In light of the previous comments, I believe the clarity has improved, the discussion has been strengthened, and all my concerns have been addressed. However, some minor revisions are still required, as detailed below:

 Case description:

-        Line 104: please, include the days of gestation for the second admission, even if only in brackets.

-        Line 134: please, use the English term (dexamethasone sodium phosphate) instead of “dexamethasonnatriumfosfaat”.

-        Line 136:  please, remove the hyphen after “clinical”.

-        Line 153:  please, remove the hyphen after “inflammatory”.

 Discussion:

-        Line 191: please, remove the hyphen after “key”.

-        Lines 218-222: in my opinion, these data should be included in the case description as part of the long-term follow-up. In the discussion, you should focus solely on commenting on their significance.

Author Response

Comment 1:

Case description:

Line 104: please, include the days of gestation for the second admission, even if only in brackets.

Thank you for your suggestion, I’ve included the days of gestation in between brackets.

Lines 104-105: Two weeks before due date (at 320 days of pregnancy), the mare was re-hospitalized at the clinic for an assisted parturition.

 Comment 2: 

Line 134: please, use the English term (dexamethasone sodium phosphate) instead of

“dexamethasonnatriumfosfaat”.

Thank you for your suggestion, I’ve changed the term.

Comment 3:

Line 136:  please, remove the hyphen after “clinical”.

Thank you for the correction, I removed the hyphen.

Comment 4:

Line 153:  please, remove the hyphen after “inflammatory”.

Thank you for the correction, I removed the hyphen.

Comment 5:

 Discussion:

 Line 191: please, remove the hyphen after “key”.

Thank you for the correction, I removed the hyphen.

Comment 6:

 Lines 218-222: in my opinion, these data should be included in the case description as part of the long-term

follow-up. In the discussion, you should focus solely on commenting on their significance.

Thank you for the suggestion, I completely agree with you. I’ve changed the section in the manuscript (Lines 146-148; lines 221-226) and also added an additional short comment in the discussion, regarding the treatment of the mare (197-200).